# Prevalence of cytopenia and its associated factors among HIV infected adults on highly active antiretroviral therapy at Mehal Meda Hospital, North Shewa Zone, Ethiopia

Angesom Gebreweld[1]*, Temesgen Fiseha[2], Nibret Girma[3], Haftay Haileslasie[1], Daniel Gebretsadik[2]

1 Department of Medical Laboratory Sciences, College of Health Sciences, Mekelle University, Mekelle, Ethiopia, 2 Department of Medical Laboratory Sciences, College of Medicine and Health Science, Wollo University, Dessie, Ethiopia, 3 Department of Laboratory, Mehal Meda Hospital, Mehal Meda, Ethiopia

* afsaha@gmail.com

## Abstract

**Data Availability Statement:** All relevant data are within the paper and its Supporting Information files.

### Background

Cytopenias affect the outcomes of highly active anti-retroviral therapy that results in higher morbidity, mortality, and impaired quality of life. The purpose of this study was to assess the prevalence of cytopenia and its associated factors among HIV infected adults on highly active antiretroviral therapy at Mehal Meda Hospital, North Shewa Zone, Ethiopia.

### Method

A cross-sectional health facility based study was conducted among 499 consecutively selected adult HIV infected patients taking HAART for at least six months from January to April 2018. The study participant's socio-demographic and clinical information was collected using a pre-tested questionnaire and reviewing of medical records by trained clinical nurses. Complete blood count and CD4 T cell count were determined by Sysmex KX-21 N and BD FACS count respectively. Bivariate and multivariate analysis was performed to identify the independently associated factors of cytopenia and prevalence ratios and their 95% confidence intervals were estimated using Poisson regression model with robust error variance to quantify the strength of statistical association. In all cases, a P value less than 0.05 was considered statistically significant.

### Result

Out of the total study participants, 39.9% had at least one form of cytopenia, 23.2% had anemia, 13.8% had leukopenia, 12.4% had thrombocytopenia, 11.62% had bi-cytopenias, and only 1% had pancytopenia. In multivariate analysis, cytopenia was independently associated with older age groups, male gender, ZDV based regimen, and CD4 count less than 200 cells/mm$^3$.

**Funding:** The author(s) received no specific funding for this work

**Competing interests:** The authors have declared that no competing interests exist.

## Conclusions

In this study, the magnitude of any cytopenia was 40% among adult HIV infected patients taking highly active antiretroviral therapy and the prevalence increased as the CD4 count decreases. Therefore, these warrant the need for monitoring hematological parameters of HIV infected patients on HAART to reduce morbidity and mortality.

## Introduction

Human immunodeficiency virus (HIV) continues to be a major global public health issue; 37.9 million people globally were living with HIV and 770,000 people died from acquired immuno-deficiency syndrome (AIDS) related illnesses at the end of 2018. Of all people living with HIV, over two-thirds live in Africa [1]. In Ethiopia, an estimated 613,000 people were living with HIV in 2017, 74% of the peoples living with HIV are from Amhara, Oromia, and Addis Ababa [2].

HIV not only targets the immune system that leads to progressive immune dysfunction but also affects the hematopoietic system of the infected individuals that result in cytopenias [3]. Cytopenias are the most common hematological complications of HIV infection and may affect any of the major blood lineages leading to anemia, thrombocytopenia, and/or leucope-nia. The causes of cytopenias in HIV infection are multifactorial, including a direct conse-quence of HIV infection, effects of medications, opportunistic infections, hepatitis B virus and hepatitis C virus co-infection, and others. The pathophysiology of cytopenias can broadly be classified as a bone marrow production defect and increased peripheral loss or destruction of blood cells [3–5]. The frequency and severity of cytopenias increase as CD4 count declines and HIV infection advances. Cytopenias can affect the outcomes of highly active antiretroviral therapy (HAART), resulting in higher morbidity, mortality, and a negative impact on the qual-ity of life [6–8]. Anemia is the most frequent cytopenia in HIV infected patients, even in patients taking HAART. It has been independently associated with accelerated HIV disease progression, mortality and decreased quality of life [9–11]. In HIV patients, the prevalence of anemia varies significantly among studies ranging from 1.3% to 95% [9]. Although treatment with HAART decreases the prevalence of severe anemia [12], the overall burden of anemia in HAART treated HIV patients remains high. In China, anemia occurs in 39.2% HIV patients receiving HAART [13]; in Europe and North America the prevalence is 26% [14]. In Ethiopia, the prevalence of anemia among HIV infected adult individuals receiving HAART ranges from 11.4% to 43% [12, 15–19].

Leucopenia is one of the hematological abnormalities encountered commonly in patients with HIV infection. Neutropenia is the most common leucopenia that occurs in 5–30% of patients with early symptomatic HIV infection and up to 70% of patients at advanced stages of AIDS [8, 20, 21]. Like other cytopenias, the cause of HIV-associated neutropenia is multifacto-rial, including a direct consequence of HIV infection, autoimmune disorders, opportunistic infections, malignancies, and drugs used to treat HIV and opportunistic infections like antire-troviral therapy (particularly zidovudine-containing regimens), cotrimoxazole, and others, which are myelotoxic [20, 22, 23].

Thrombocytopenia is also commonly observed cytopenia in HIV-infected patients. It may occur in 4.1–40% of HIV patients and the prevalence and severity of thrombocytopenia increases as the disease stage advances [3, 20]. The causes of thrombocytopenia include immune-mediated platelet destruction, impaired megakaryocytosis/direct infection of

megakaryocytes, hypersplenism, opportunistic infections, malignancy, and myelosuppression effects of medications [20, 24].

Cytopenia remains problematic in resource limited countries like Ethiopia. The diagnosis of cytopenia and its underlying mechanisms during antiretroviral therapy ensure optimal management of patients with HIV disease [25]. However, in our setting, there is scarce information on the magnitude and factors associated with cytopenias in patients under HAART. Therefore, the purpose of this study was to assess prevalence of cytopenia and its associated factors among HIV infected patients on HAART at Mehal Meda Hospital, North Shewa Zone, Ethiopia.

## Materials and methods

### Study design, setting and population

A cross-sectional study was conducted to assess the prevalence of cytopenia and its associated factors among HIV infected patients on HAART at Mehal Meda Hospital from January to April 2018. The Hospital is found in Mehal Meda town in Menz Gera Midir Woreda North Shewa Zone of the Amhara Region, Ethiopia. Mehal Meda town is about 280 km away from the capital city Addis Ababa and 148 km from Debre Brhan city. Mehal Meda Hospital provides comprehensive health care services including HIV/AIDS diagnosis, treatment and monitoring for a catchment population of more than 360,000.

A total of 499 consecutively selected adult HIV infected patients taking HAART for at least six months were enrolled from ART follow-up unit of Mehal Meda Hospital. The sample size was calculated using single proportion formula with an assumption of a proportion of cytopenia among HIV infected patients taking HAART of 50% since there is no study conducted in the area. The assumptions also include 95% confidence interval, 5% marginal error and 10% non-response rate. The calculated sample size become 422 but we included 499 subjects into the study. HIV-infected patients with known hematological disorders, pregnancy, postpartum period, blood transfusion (within three months) and patients younger than 18 years were excluded from the study.

### Data collection procedure

The study participant's socio-demographic and clinical information was collected using a pretested questionnaire and by reviewing of medical records by trained clinical nurses. Height and weight of the participants were measured by the data collectors.

Venous blood specimen (about 4 ml) in EDTA vacutainer tube was collected from each study participants by a senior laboratory professional for complete blood count and CD4 T cell count determination. Sysmex KX-21N (Sysmex corporation kobe, Japan) automated hematological analyzer were used to determine complete blood count (total white blood cell count (WBC), red blood cell count, platelet (PLT) count, and hemoglobin concentration level) and BD FACS count (Becton Dickenson and Company, California, USA) machine was used to measure CD4 T cell count of the study participants. Standard operating procedures and manufacturers' instructions were strictly followed in each laboratory procedures to maintain quality of the laboratory results. Expiry date of reagents were checked and quality control materials were run along with patient samples to check precision of the instruments and accuracy of the results.

### Operational definitions

Anemia was defined based on the World Health Organization (WHO) criteria: hemoglobin (Hgb) concentration <13 g/dl for males (15 years of age and above) and < 12 g/dl for females

(non-pregnant women). Anemia was also categorized using the WHO classification into mild anemia (for men: Hgb level from 11.0–12.9 g/dl, for non-pregnant women: Hgb level from 11.0–11.9 g/dl), moderate anemia (Hgb level from 8.0–10.9 g/dl for both sexes), and severe anemia (Hgb lower than 8.0 g/dl for both sexes) [26].

As anemia, there is no generally accepted cut-offs for leucopenia and thrombocytopenia; we defined them as used in other studies [19, 27]. Leucopenia was defined as a total WBC $< 4.0 \times 103/\mu L$, whereas thrombocytopenia was defined as a total platelet count $< 150 \times 103/\mu L$. The platelet counts from 100 to150 $\times 103/\mu L$, 50 to 100 $\times 103/\mu L$, and less than $50 \times 103/\mu L$ were considered as mild thrombocytopenia, moderate thrombocytopenia, and severe thrombocytopenia, respectively [28].

Any cytopenia was defined as presence of at least one form of cytopenia (anemia, thrombocytopenia, or leucopenia), bi-cytopenia was defined as presence of two forms of cytopenias (anemia, thrombocytopenia and leucopenia) and pancytopenia was defined as having all three forms of cytopenia simultaneously on the study participants.

## Statistical analysis

All collected data were entered in to "Epi Data version 3.1" and exported to Stata® 12.0 (StataCorp, College Station, Texas) statistical software for analysis. Continuous variables were expressed as median and inter-quartile range (IQR) and categorical variables were reported as frequencies and percentages. Chi-square (×2) test for categorical variables was performed to assess their relation with the outcomes. Bivariate and multivariate analysis was performed to identify the independently associated factors of cytopenias and prevalence ratios and their 95% confidence intervals were estimated using Poisson regression model with robust error variance to quantify the strength of statistical association. Variables with p < 0.20 at bivariate analysis were included in the multivariable analysis. In all cases, a P value less than 0.05 was considered statistically significant.

## Ethical considerations

Ethical approval was obtained after the study protocol was approved by Research and Ethics Review Committee of the Department of Medical Laboratory Sciences, Wollo University and a letter of permission to conduct the study was obtained from Mehal Meda Hospital. Individual written informed consent was obtained after the purpose and importance of the study were explained. To ensure confidentiality, participant's identifiers were removed and only code numbers were used throughout the study. Any abnormal test results of participants were reported to the concerned body in the ART clinic of the hospital.

## Result

### Sociodemographic characteristics of study participants

A total of 499 HIV positive patients taking ART were included in this study. The median age of the study participants was 36 years (Interquartile Range: 29–43) and most of the study participants (n = 163; 32.7%) were within 30–39 years of age. Of the total 499 study participants, 260 (52.1%) were females, 301 (60.3%) were from urban setting, 394 (79.0%) had primary education and below, 240 (48.1%) were married and 163 (32.7%) were employed (Table 1).

### Clinical and laboratory characteristics of the study participants

Among the study participants, 295 (59.1%) were in WHO clinical stage I and 226 (45.3%) were taking 1e (TDF-3TC-EFV) ART regimen. Majority of the study participants (n = 329; 65.9%)

**Table 1.  Sociodemographic characteristics of HIV positive patients taking ART at Mahal Meda Hospital, North Shewa, Ethiopia, 2018 (n = 499).**

| Variables | Category | Frequency (n = 499) | Percentage (%) |
|---|---|---:|---:|
| Age (in years) | 18–29 | 138 | 27.7 |
|  | 30–39 | 163 | 32.7 |
|  | 40–49 | 149 | 29.9 |
|  | > = 50 | 49 | 9.8 |
| Sex | Male | 239 | 47.9 |
|  | Female | 260 | 52.1 |
| Residence | Urban | 301 | 60.3 |
|  | Rural | 198 | 39.7 |
| Educational status | Primary school and below | 394 | 79.0 |
|  | High school and above | 105 | 21.0 |
| Occupational status | Employed | 163 | 32.7 |
|  | Unemployed | 336 | 67.3 |
| Marital status | Married | 240 | 48.1 |
|  | Unmarried | 259 | 51.9 |
| Family size | Three and below | 273 | 54.7 |
|  | Above three | 226 | 45.3 |

were used the ART regimen with in duration group of 12–59 months with median of 31.0 months (interquartile range: 17.0–60.0 months). One hundred twenty nine (25.9%) of the study participants had opportunistic infection (tuberculosis; 24 (4.8%), pneumonia; 34 (6.8%), oral candidiasis; 7 (1.4%), and diarrhea; 64 (12.8%)) and 30 (6%) of the study participants had chronic disease (diabetic mellitus 9 (1.8%) and hypertension 21 (4.2%)). The median body mass index of the study participants was 19.65 Kg/m$^2$ (interquartile range: 17.96–21.33 Kg/m2) and 150 (30.1%) study participants were under weight (BMI<18.5 Kg/m$^2$). Majority of the respondents 245 (49.1%) had CD4 cell count greater than 500 cell/μL while only 70 (14%) of the respondents had less than 200 cell/μL. The median CD4 count, hemoglobin concentration, white blood cell count and platelet count was 482.0 cell/μL (interquartile range: 290.0–669.0 cell/μL), 14.20 g/dL (interquartile range: 12.60–15.60 g/dL), 6.40 x10$^3$/μL (interquartile range: 4.80–7.90 x10$^3$/μL), and 242.0 x10$^3$/μL (interquartile range: 194.0–301.0 x10$^3$/μL), respectively (Table 2).

## Prevalence and potential risk factors of cytopenias

Out of the total study participants, 39.9% had any cytopenias, 23.2% had anemia, 13.8% had leukopenia, 12.4% had thrombocytopenia, 11.62% had bi-cytopenias, and only 1% had Pancytopenia. The most frequent bi-cytopenias was anemia and thrombocytopenia combination (4.2%). Of the anemic study subjects (n = 116), 8.62% had severe anemia, 54.31% had moderate anemia, and 37.07% had mild anemia. Majority of the thrombocytopenic study participants had mild thrombocytopenia (87.1%) (Table 3).

The prevalence of any cytopenia (at least one form of cytopenia) was significantly associated with male patients, older age groups, rural dwellers, being unmarried by marital status, WHO clinical stage II, ZDV based regimen, taking ART regimen below 60 months, presence of opportunistic infection, and CD4 count less than 200 cells/mm$^3$ and 200–499 cells/mm$^3$on the bivariate analysis.

Using multivariate logistic regression analysis, cytopenia was independently associated with male gender (PR = 1.34, 95% CI: 1.07–1.67, P = .011), unmarried (PR = 1.36, 95% CI: 1.09–1.69, P = 0.007), ZDV based regimen (PR = 1.56, 95% CI: 1.24–1.96, P<0.001), taking ART

**Table 2. Clinical and laboratory characteristics of HIV infected patients on HAART at Mahal-Meda Hospitals, Northern Shewa Zone, Ethiopia, 2018 (n = 499).**

| Variables | Category | Frequency (n = 499) | Percentage (%) | Median (IQR) |
|---|---|---|---|---|
| WHO clinical stages | Stage I | 295 | 59.1 | - |
| | Stage II | 145 | 29.1 | - |
| | Stage III | 53 | 10.6 | - |
| | Stage IV | 6 | 1.2 | - |
| Types of ART regimens | 1C | 163 | 32.7 | - |
| | 1D | 89 | 17.8 | - |
| | 1E | 226 | 45.3 | - |
| | 1F | 10 | 2.0 | - |
| | 1H | 11 | 2.2 | - |
| Duration of ART regiment | <12 months | 46 | 9.2 | - |
| | 12–59 months | 329 | 65.9 | - |
| | > = 60 months | 124 | 24.8 | - |
| Opportunistic infection | TB | 24 | 4.8 | - |
| | Pneumonia | 34 | 6.8 | - |
| | Oral candidiasis | 7 | 1.4 | - |
| | Diarrhea | 64 | 12.8 | - |
| | No | 370 | 74.1 | - |
| Chronic illness | Diabetic Mellitus | 9 | 1.8 | - |
| | Hypertension | 21 | 4.2 | - |
| | NO | 469 | 94.0 | - |
| BMI category | 18.5–24.9 | 330 | 66.1 | - |
| | <18 | 150 | 30.1 | - |
| | > = 25 | 19 | 3.8 | - |
| CD4 count (cells/μL) | <200 | 71 | 14.2 | - |
| | 200–499 | 183 | 36.7 | - |
| | > = 500 | 245 | 49.1 | - |
| Duration in months | - | - | 31.0 (17.0–60.0) |
| BMI Kg/m2 | - | - | 19.65 (17.96–21.33) |
| CD4 count (cells/μL) | - | - | 482.0 (290.0–669.0) |
| White blood cell count (x10³/μL) | - | - | 6.40 (4.80–7.90) |
| Hemoglobin concentration (g/dL) | - | - | 14.20 (12.60–15.60) |
| Platelet count (x10³/μL) | - | - | 242.0 (194.0–301.0) |

WHO = World Health Organization, IQR = interquartile range, 1c = ZDV -3TC-NVP, 1d = ZDV-3TC-EFV, 1e = TDF-3TC-EFV, 1f = TDF-3TC-NVP, 1h = ABC+3TC +EFV, ZDV = zidovudine, 3TC = lamuvidine, EFV = efaverenz, NVP = neverapine, TDF = tenofovir, ABC = abacavir, BMI = body mass index.

regiment for 12–60 months (PR = 1.54, 95% CI: 1.16–2.06, P = 0.003), CD4 count less than 200 cells/mm³ (PR = 2.06, 95% CI: 1.54–2.75, P<0.001) and 200–499 cells/mm³ (PR = 1.36, 95% CI: 1.06–1.74, P = 0.015) (Table 4).

The prevalence of anemia was significantly higher in males than females (28% vs18.8%, P = 0.016). Anemia was also higher in HIV patients with older age group, rural residents, WHO clinical stage III and IV, ZDV based ART regimen, and opportunistic infection. The prevalence of anemia was decreased as ART regiment usage becomes longer. However, the prevalence of anemia increased as the CD4 T cell count decreased (Table 5).

In multivariable analysis, the prevalence ratio of having anemia in older age group (≥50 years) was 1.4 (PR = 1.39, 95% CI: 1.05–2.32, P = 0.027) compared to younger age group (18–29 years). The prevalence ratio of anemia was twice in rural residents (PR = 2.05, 95% CI:

**Table 3. Prevalence of cytopenias among HIV infected patients on HAART at Mahal-Meda Hospitals, Northern Shewa Zone, Ethiopia, 2018 (n = 499).**

| Hematological abnormalities | Frequency (%) |
| --- | --- |
| **Any cytopenia** | 199 (39.9) |
| Anemia | 116 (23.2) |
| Leucopenia | 69 (13.8) |
| Thrombocytopenia | 62 (12.4) |
| **Bi-cytopenia** | |
| Anemia and leucopenia | 18 (3.6) |
| Anemia and thrombocytopenia | 21 (4.2) |
| Thrombocytopenia and leucopenia | 14 (2.8) |
| **Pancytopenia** | 5 (1.0) |
| **Anemia severity (n = 116)** | |
| Mild | 43 (37.07) |
| Moderate | 63 (54.31) |
| Severe | 10 (8.62) |
| **Thrombocytopenia severity (n = 62)** | |
| Mild | 54 (87.10) |
| Moderate | 4 (6.45) |
| Severe | 4 (6.45) |

Mild anemia = (Hgb level: 11.0–12.9 g/dl for men, 11.0–11.9 g/dl for women), moderate anemia (Hgb level 8.0–10.9 g/dl for both sexes), severe anemia (Hgb level <8.0 g/dl for both sexes), mild thrombocytopenia (platelet counts: 100 to150 × 103/μL), moderate thrombocytopenia (platelet counts: 50 to 100 × 103/μL), severe thrombocytopenia (< 50 × 103/μL).

1.48–2.84, P<0.001) compared to urban residents, 1.5 (PR = 1.46, 95% CI: 1.04–2.06, P = 0.028) in unmarried study participants compared to married participants, and 1.5 times higher in WHO clinical stage II (PR = 1.51, 95% CI: 1.08–2.10, P = 0.016) compare to participants in WHO clinical stage I. Anemia was also independently associated with 12–60 months duration of ART regimen (PR = 1.93, 95% CI: 1.24–3.02, P = 0.004) and opportunistic infection (PR = 1.69, 95%CI: 1.13–2.23, P = 0.007) as summarized in Table 5.

The prevalence of leucopenia among males and females were 13.4% and 14.2%, respectively but the difference were not statistically significant (P = 0.786). Leucopenia prevalence progressively increased with age: 8.0% for age 18–29 years, 10.4% for 30–39 years, 18.8% for 40–49 years and 26.5% for >50 years (P = 0.002). Leucopenia had decreased as CD4+ T-cell counts increased: CD4 count <200 cells/mm$^3$ (38.0%), CD4 count from 200–499 cells/mm$^3$ (15.8%), and CD4 count >500 cell/mm$^3$ (5.3%) (P<0.001). Using a multivariate analysis, leucopenia was independently associated with age category 40–49 years (PR = 2.06, 95% CI: 1.12–3.79, P = 0.019), >50 years (PR = 2.40, 95% CI: 1.26–4.61, P = 0.008), ZDV based regimen (PR = 1.86, 95% CI: 1.15–3.00, P = 0.004), CD4 count less than 200 cells/mm$^3$ (PR = 5.99, 95% CI: 3.18–11.29, P<0.001), and CD4 count from 200–499 cells/mm$^3$ (PR = 2.91, 95% CI: 1.56–5.42, P = 0.001) (Table 6).

In this study, the overall prevalence of thrombocytopenia was 12.4%. The prevalence of thrombocytopenia among males (17.6%) were higher than females (7.7%) and the difference were statistically significant (P = .001). In bivariate analysis, thrombocytopenia was associated with age group, male gender, occupational status, marital status, WHO clinical stage III and IV, AZT based ART regimen, CD4 count less than 200 cell/mm$^3$. In multivariate analysis, thrombocytopenia was independently associated with age group 40–49 years (PR = 0.33, 95%

**Table 4. Factors associated with cytopenia among HIV infected patients on HAART at Mahal-Meda Hospitals, Northern Shewa Zone, Ethiopia, 2018 (n = 499).**

| Variable | Any Cytopenia | | Bivariable | P-value | Multivariable | P-value |
|---|---|---|---|---|---|---|
| | No, n (%) | Yes, n (%) | PR (95% CI) | | PR (95% CI) | |
| **Age group (in years)** | | | | | | |
| 18–29 | 83 (60.1) | 55 (39.9) | 1 | | 1 | |
| 30–39 | 103 (63.2) | 60 (36.8) | 0.92 (0.69–1.23) | 0.587 | 1.02 (0.75–1.37) | 0.918 |
| 40–49 | 94 (63.1) | 55 (36.9) | 0.93 (0.69–1.24) | 0.608 | 0.93 (0.68–1.28) | 0.667 |
| > = 50 | 20 (40.8) | 29 (59.2) | **1.48 (1.09–2.03)** | **0.012** | 1.33 (0.93–1.90) | 0.116 |
| **Sex** | | | | | | |
| Male | 124 (51.9) | 115 (48.1) | **1.49 (1.19–1.86)** | **<0.001** | **1.34 (1.07–1.67)** | 0.011 |
| Female | 176 (67.7) | 84 (32.3) | 1 | | 1 | |
| **Residence** | | | | | | |
| Urban | 193 (64.1) | 108 (35.9) | 1 | | 1 | |
| Rural | 107 (54.0) | 91 (46.0) | **1.28 (1.04–1.59)** | **0.023** | 1.22 (0.98–1.52) | 0.076 |
| **Educational status** | | | | | | |
| Primary school and below | 242 (61.4) | 152 (38.6) | 0.86 (0.67–1.10) | 0.237 | | |
| High school and above | 58 (55.2) | 47 (44.8) | 1 | | | |
| **Occupational status** | | | | | | |
| Employed | 105 (64.4) | 58 (35.6) | 1 | | 1 | |
| Un employed | 195 (58.0) | 141 (42.0) | 1.18 (0.93–1.50) | 0.181 | 1.01 (0.80–1.28) | 0.914 |
| **Marital status** | | | | | | |
| Married | 164 (68.3) | 76 (31.7) | 1 | | 1 | |
| Unmarried | 136 (52.5) | 123 (47.5) | **1.50 (1.19–1.88)** | **<0.001** | **1.36 (1.09–1.69)** | 0.007 |
| **WHO clinical stage** | | | | | | |
| stage I | 191 (64.7) | 104 (35.3) | 1 | | 1 | |
| Stage II | 77 (53.1) | 68 (46.9) | **1.33 (1.05–1.68)** | **0.016** | 1.13 (0.90–1.41) | 0.295 |
| Stage III and IV | 32 (54.2) | 27 (45.8) | 1.29 (0.95–1.78) | 0.108 | 0.81 (0.58–1.13) | 0.221 |
| **Regimen** | | | | | | |
| ZDV based | 123 (48.8) | 129 (51.2) | **1.81 (1.43–2.28)** | **<0.001** | **1.55 (1.24–1.96)** | **<0.001** |
| Non ZDV based | 177 (71.7) | 70 (28.3) | 1 | | 1 | |
| **Duration** | | | | | | |
| <12 months | 24 (52.2) | 22 (47.8) | **1.60 (1.07–2.40)** | 0.022 | 1.49 (0.97–2.32) | 0.071 |
| 12–60 months | 189 (57.4) | 140 (42.6) | **1.43 (1.06–1.92)** | 0.019 | **1.54 (1.16–2.06)** | 0.003 |
| >60 months | 87 (70.2) | 37 (29.8) | 1 | | 1 | |
| **Opportunistic infection** | | | | | | |
| Yes | 59 (45.7) | 70 (54.3) | **1.56 (1.26–1.92)** | **<0.001** | 1.05 (0.84–1.32) | 0.655 |
| No | 241 (65.1) | 129 (34.9) | 1 | | 1 | |
| **Chronic disease** | | | | | | |
| Yes | 14 (46.7) | 16 (53.3) | 1.37 (0.96–1.95) | 0.083 | 0.97 (0.65–1.44) | 0.878 |
| No | 286 (61.0) | 183 (39.0) | 1 | | 1 | |
| **BMI category** | | | | | | |
| 18.5–24.9 | 198 (59.8) | 133 (40.2) | 1 | | | |
| <18 | 88 (59.1) | 61 (40.9) | 1.02 (0.81–1.29) | 0.875 | | |
| > = 25 | 14 (73.7) | 5 (26.3) | 0.65 (0.31–1.41) | 0.277 | | |
| **CD4 count category** | | | | | | |
| <200 | 26 (36.6) | 45 (63.4) | **2.19 (1.68–2.85)** | **<0.001** | **2.06 (1.54–2.75)** | **<0.001** |
| 200–499 | 100 (54.6) | 83 (45.4) | **1.56 (1.22–2.01)** | 0.001 | **1.36 (1.06–1.74)** | 0.015 |
| > = 500 | 174 (71.0) | 71 (29.0) | 1 | | 1 | |

WHO = World Health Organization, ZDV = zidovudine, PR = Prevalence ratio, CI = Confidence interval, 1.00 = reference group.

**Table 5. Factors associated with anemia among HIV infected patients on HAART at Mahal-Meda Hospitals, Northern Shewa Zone, Ethiopia, 2018 (n = 499).**

| Variable | Anemia | | Bivariable | P-value | Multivariable | P-value |
|---|---|---|---|---|---|---|
| | No, n (%) | Yes, n (%) | PR (95% CI) | | PR (95% CI) | |
| Age group (in years) | | | | | | |
| 18–29 | 106 (76.8) | 32 (23.2) | 1 | | 1 | |
| 30–39 | 126 (77.3) | 37 (22.7) | 0.98 (0.65–1.48) | 0.920 | 0.98 (0.62–1.55) | 0.926 |
| 40–49 | 121 (81.2) | 28 (18.8) | 0.81 (0.52–1.27) | 0.362 | 0.84 (0.52–1.35) | 0.463 |
| > = 50 | 30 (61.2) | 19 (38.8) | **1.68 (1.05–2.66)** | **0.030** | 1.39 (1.05–2.32) | 0.027 |
| **Sex** | | | | | | |
| Male | 172 (72.0) | 67 (28.0) | **1.49 (1.08–2.06)** | **0.016** | 1.22 (0.89–1.67) | 0.210 |
| Female | 211 (81.2) | 49 (18.8) | 1 | | 1 | |
| **Residence** | | | | | | |
| Urban | 253 (84.1) | 48 (15.9) | 1 | | 1 | |
| Rural | 130 (65.7) | 68 (34.3) | **2.15 (1.56–2.98)** | **<0.001** | **2.05 (1.48–2.84)** | **<0.001** |
| **Educational status** | | | | | | |
| Primary school and below | 305 (77.4) | 89 (22.6) | 0.88 (0.60–1.28) | 0.496 | | |
| High school and above | 78 (74.3) | 27 (25.7) | 1 | | | |
| **Occupational status** | | | | | | |
| Employed | 131 (80.4) | 32 (19.6) | 1 | | 1 | |
| Un employed | 252 (75.0) | 84 (25.0) | 1.27 (0.89–1.83) | 0.191 | 0.88 (0.61–1.28) | 0.516 |
| **Marital Status** | | | | | | |
| Married | 199 (82.9) | 41 (17.1) | 1 | | 1 | |
| Unmarried | 184 (71.0) | 75 (29.0) | **1.69 (1.21–2.38)** | 0.002 | **1.46 (1.04–2.06)** | 0.028 |
| **WHO clinical stage** | | | | | | |
| Stage I | 243 (82.4) | 52 (17.6) | 1 | | 1 | |
| Stage II | 100 (69.0) | 45 (31.0) | **1.76 (1.25–2.50)** | **0.001** | **1.51 (1.08–2.10)** | 0.016 |
| Stage III and IV | 40 (67.8) | 19 (32.2) | **1.83 (1.17–2.85)** | **0.008** | 1.12 (0.69–1.79) | 0.649 |
| **Regimen** | | | | | | |
| ZDV based | 178 (70.6) | 74 (29.4) | **1.73 (1.24–2.42)** | 0.001 | 1.25 (0.89–1.74) | 0.187 |
| Non ZDV based | 205 (83.0) | 42 (17.0) | 1 | | 1 | |
| **Duration** | | | | | | |
| <12 months | 33 (71.7) | 13 (28.3) | **2.08 (1.09–3.90)** | **0.026** | **2.03 (1.01–4.07)** | 0.046 |
| 12–60 months | 243 (73.9) | 86 (26.1) | **1.91 (1.18–3.07)** | **0.008** | **1.93 (1.24–3.02)** | 0.004 |
| >60 months | 107 (86.3) | 17 (13.7) | 1 | | 1 | |
| **Opportunistic infection** | | | | | | |
| Yes | 76 (58.9) | 53 (41.1) | **2.41 (1.78–3.28)** | **<0.001** | **1.69 (1.13–2.23)** | 0.007 |
| No | 307 (83.0) | 63 (17.0) | 1 | | 1 | |
| **Chronic disease** | | | | | | |
| Yes | 19 (63.3) | 11 (36.7) | 1.64 (0.99–2.70) | 0.053 | 1.05 (0.59–1.88) | 0.859 |
| No | 364 (77.6) | 105 (22.4) | 1 | | 1 | |
| **BMI category** | | | | | | |
| 18.5–24.9 | 258 (77.9) | 73 (22.1) | 1 | | | |
| <18 | 111 (74.5) | 38 (25.5) | 1.16 (0.82–1.63) | 0.404 | | |
| > = 25 | 14 (73.7) | 5 (26.3) | 1.19 (0.54–2.61) | 0.657 | | |
| **CD4 count category** | | | | | | |
| <200 | 47 (66.2) | 24 (33.8) | **1.76 (1.16–2.67)** | **0.007** | **1.53 (1.01–2.31)** | 0.045 |
| 200–499 | 138 (75.4) | 45 (24.6) | 1.28 (0.89–1.84) | 0.178 | 0.91 (0.65–1.28) | 0.594 |
| > = 500 | 198 (80.8) | 47 (19.2) | 1 | | 1 | |

WHO = World Health Organization, ZDV = Zidovudine, PR = Prevalence ratio, CI = Confidence interval, 1.00 = Reference group.

**Table 6. Factors associated with leucopenia among HIV infected patients on HAART at Mahal-Meda Hospital, Northern Shewa Zone, Ethiopia, 2018 (n = 499).**

| Variable | Leucopenia | | Bivariable | P value | Multivariable | P-value |
|---|---|---|---|---|---|---|
| | No, n (%) | Yes, n (%) | PR (95% CI) | | PR (95% CI) | |
| **Age group (in years)** | | | | | | |
| 18–29 | 127 (92.0) | 11 (8.0) | 1 | | 1 | |
| 30–39 | 146 (89.6) | 17 (10.4) | 1.31 (.63–2.70) | 0.467 | 1.61 (0.81–3.19) | 0.172 |
| 40–49 | 121 (81.2) | 28 (18.8) | **2.36 (1.22–4.55)** | **0.011** | **2.06 (1.12–3.79)** | 0.019 |
| > = 50 | 36 (73.5) | 13 (26.5) | **3.33 (1.59–6.94)** | **0.001** | **2.40 (1.26–4.61)** | 0.008 |
| **Sex** | | | | | | |
| Male | 207 (86.6) | 32 (13.4) | **1** | | | |
| Female | 223 (85,8) | 37 (14.2) | 1.06 (0.68–1.65) | 0.786 | | |
| **Residence** | | | | | | |
| Urban | 257 (85.4) | 44 (14.6) | 1 | | | |
| Rural | 173 (87.4) | 25 (12.6) | 0.86 (0.55–1.36) | 0.530 | | |
| **Educational status** | | | | | | |
| Primary school and below | 344 (87.3) | 50 (12.7) | 0.70 (0.43–1.14) | 0.150 | 0.70 (0.44–1.12) | 0.142 |
| High school and above | 86 (81.9) | 19 (18.1) | 1 | | 1 | |
| **Occupational status** | | | | | | |
| Employed | 138 (84.7) | 25 (15.3) | 1 | | | |
| Un employed | 292 (86.9) | 44 (13.1) | 0.85 (0.54–1.35) | 0.495 | | |
| **Marital status** | | | | | | |
| Married | 205 (85.4) | 35 (14.6) | 1 | | | |
| Unmarried | 225 (86.9) | 34 (13.1) | 0.90 (0.58–1.40) | 0.638 | | |
| **WHO clinical stage** | | | | | | |
| stage I | 254 (86.1) | 41 (13.9) | 1 | | 1 | |
| Stage 2 | 133 (91.7) | 12 (8.3) | 0.59 (0.32–1.09) | 0.097 | **0.42 (0.23–0.76)** | 0.004 |
| Stage 3and 4 | 43 (72.9) | 16 (27.1) | **1.95 (1.18–3.23)** | **0.010** | 1.34 (0.78–2.29) | 0.727 |
| **Regimen** | | | | | | |
| ZDV based | 206 (81.7) | 46 (18.3) | **1.96 (1.23–3.13)** | 0.005 | **1.86 (1.15–3.00)** | **0.011** |
| Non ZDV based | 224 (90.7) | 23 (9.3) | 1 | | | |
| **Duration** | | | | | | |
| <12 months | 38 (82.6) | 8 (17.4) | **1** | | | |
| 12–60 months | 290 (88.1) | 39 (11.9) | 0.681 (0.34–1.37) | 0.280 | | |
| >60 months | 102 (82.3) | 22 (17.7) | 1.02 (0.49–2.13) | 0.958 | | |
| **Opportunistic infection** | | | | | | |
| Yes | 113 (87.6) | 16 (12.4) | 0.86 (0.51–1.46) | 0.589 | | |
| No | 317 (85.7) | 53 (14.3) | 1 | | | |
| **Chronic disease** | | | | | | |
| Yes | 26 (86.7) | 4 (13.3) | 0.96 (0.38–2.46) | 0.936 | | |
| NO | 404 (86.1) | 65 (13.9) | 1 | | | |
| **BMI category** | | | | | | |
| 18.5–24.9 | 281 (84.9) | 50 (15.1) | 1 | | | |
| <18 | 130 (87.2) | 19 (12.8) | 0.84 (0.52–1.38) | 0.500 | | |
| > = 25 | 19 (100) | 0 (0) | 0.00 (0.00) | 0.985 | | |
| **CD4 count category** | | | | | | |
| <200 | 44 (62.0) | 27 (38.0) | **7.16 (3.91–13.15)** | **<0.001** | **5.99 (3.18–11.29)** | **<0.001** |
| 200–499 | 154 (84.2) | 29 (15.8) | **2.99 (1.59–5.59)** | **0.001** | **2.91 (1.56–5.42)** | **<0.001** |
| > = 500 | 232 (94.7) | 13 (5.3) | 1 | | 1 | |

WHO = World Health Organization, ZDV = Zidovudine, PR = Prevalence ratio, CI = Confidence interval, 1.00 = Reference group.

CI: 0.16–0.70, P = 0.004), male participants (PR = 2.36, 95% CI: 1.37–4.07, P = 0.002), unmarried (PR = 2.36, 95% CI: 1.33–4.20, P = 0.003), and CD4 count less than 200 cell/mm$^3$ (PR = 2.91, 95% CI: 1.58–5.35, P = 0.001) (Table 7).

## Discussion

This study assessed the prevalence and associated factors of cytopenias among adult HIV infected patients on HAART at Mehal Meda Hospital, North Shewa Zone, Ethiopia. The overall prevalence of at least one form of cytopenia (presence of anemia, thrombocytopenia or leukopenia) was 39.9% (95% CI;35.5% - 43.5%) and it was independently associated with older age groups, male gender, unmarried marital status, ZDV based ART regimen, taking ART regiment for 12–59 months, and CD4 count less than 200 cells/mm$^3$. Prevalence of cytopenia in this study was higher than study done in Beijing Ditan Hospital, China [29]. These differences could be due to different cut-off values used to define the cytopenias, study population, socioeconomic status and dietary habits of study participants.

In this study, the most frequent type of cytopenia were anemia and its prevalence was 23.2% (95% CI; 19.4% - 26.8%) which was in agreement with other findings reported in Gondar (22.2%) [17], Northeastern Nigeria (24.3%) [30], and Kaduna State, Nigeria (23%) [31]. However, the prevalence was lower compared to studies conducted in Debre Tabor (29.9%) [16], Tikur Anbessa Specialized Hospital, Addis Ababa (34.6%) [18], Jimma University Specialized Hospital, Jimma (43.1) [19], South West Region of Cameroon (58.6%) [32], Benin City, Nigeria (51.15%) [33], and Brazil (37.5%) [34]. Our finding was higher than studies conducted in Gondar University Hospital (11.7%) [15], Black Lion Specialized Hospital, (11.4%) [12], and Zewditu Memorial Hospital (14.3%) [35]. The reasons for the observed differences in prevalence of anemia might be due to the difference in study population, socioeconomic status and dietary habits of study participants, sample size, and difference in the definition of anemia.

Of the anemic study subjects, about 9% had severe anemia, 54% had moderate anemia, and 37% had mild anemia. The predominance of moderate type of anemia in the current study is in line with a study conducted at Tikur Anbessa Specialized Hospital, Addis Ababa [18] but it deviates from the findings reported in Wolita Sodo University, Sodo [36], Black Lion Hospital, Addis Ababa [12], Debre Tabor [37], and rural China [13], which reported high rate of mild anemia.

In the present study, the prevalence of anemia was significantly higher in males than females which is consistent with other studies [13, 33, 35]; this might be due to the difference in the definition of anemia. However, many studies reported a high prevalence of anemia in females than male HIV infected patients [18, 37–40]. Similar to other studies findings [12, 13, 19, 39], we found that the prevalence of anemia was significantly higher as the CD4 T cell count of the study participants decreased and WHO clinical stage advanced. This might be due to high HIV infection that leads to compromised immune system and disrupts normal hematopoiesis by cytokine dysregulation [3, 4]. Anemia also independently associated with older age group (≥50 years) and opportunistic infections in this study, this might be associated with immunosuppression. The prevalence of anemia was decreased with longer ART regiment usage and this is attributed to the positive effect of ART on the differentiation and survival of erythrocytes. The finding is supported by other studies [12, 32].

Leukopenia was found in 13.8% (95% CI; 11.1% - 17.1%) of the study participants, this finding was supported by a study conducted in Jimma, Ethiopia (12.3%) [19]. However, the prevalence was lower than studies in Gonder, Ethiopia (35.9%) [15], South West Region of Cameroon (20%) [32], and Karnataka, India (35%) [41] and higher than studies in Ghana

**Table 7. Factors associated with thrombocytopenia among HIV infected patients on HAART at Mahal-Meda Hospitals, Northern Shewa Zone, Ethiopia, 2018 (n = 499).**

| Variable | Thrombocytopenia | | Bi-variable | P value | Multivariable | P-value |
|---|---|---|---|---|---|---|
| | No, n (%) | Yes, n (%) | PR (95% CI) | | PR (95% CI) | |
| **Age group (in years)** | | | | | | |
| 18–29 | 112 (81.2) | 26 (18.8) | 1 | | 1 | |
| 30–39 | 147 (90.2) | 16 (9.8) | 0.52 (0.29-.93) | 0.028 | **0. 69 (0.38–1.23)** | 0.208 |
| 40–49 | 140 (94.0) | 9 (6.0) | **0.32 (0.16–0.66)** | **0.002** | **0.33 (0.16-.70)** | 0.004 |
| > = 50 | 38 (77.6) | 11 (22.4) | 1.19 (0.64–2.23) | 0.583 | 0.91 (0.45–1.84) | 0.801 |
| Sex | | | | | | |
| Male | 197 (82.4) | 42 (17.6) | **2.28 (1.38–3.78)** | **0.001** | **2.36 (1.37–4.07)** | **.002** |
| Female | 240 (92.3) | 20 (7.7) | 1 | | 1 | |
| **Residence** | | | | | | |
| Urban | 260 (86.4) | 41 (13.6) | 1 | | | |
| Rural | 177 (89.4) | 21 (10.6) | 0.78 (0.47–1.28) | 0.322 | | |
| **Educational status** | | | | | | |
| Primary school and below | 352 (89.3) | 42 (10.7) | 0.56 (0.34–0.91) | 0.020 | 0.82 (0.54–1.25) | 0.356 |
| High school and above | 85 (81.0) | 20 (19.0) | 1 | | 1 | |
| Occupational status | | | | | | |
| Employed | 151 (92.6) | 12 (7.4) | 1 | | 1 | |
| Un employed | 286 (85.1) | 50 (14.9) | **2.02 (1.11–3.69)** | 0.022 | 1.57 (0.85–2.88) | 0.147 |
| Marital status | | | | | | |
| Married | 226 (94.2) | 14 (5.8) | 1 | | 1 | |
| Unmarried | 211 (81.5) | 48 (18.5) | **3.18 (1.79–5.61)** | <0.001 | **2.36 (1.33–4.21)** | 0.003 |
| **WHO clinical stage** | | | | | | |
| Stage I | 265 (89.8) | 30 (10.2) | 1 | | 1 | |
| Stage 2 | 125 (86.2) | 20 (13.8) | 1.36 (0.79–2.31) | 0.260 | 1.15 (0.68–1.96) | 0.595 |
| Stage 3and 4 | 47 (79.7) | 12 (20.3) | **2.00 (1.09–3.68)** | **0.026** | 1.17 (0.62–2.21) | 0.624 |
| **Regimen** | | | | | | |
| ZDV based | 210 (83.3) | 42 (16.7) | 2.06 (1.24–3.40) | 0.005 | 1.40 (0.82–2.33) | 0.227 |
| Non ZDV based | 227 (91.9) | 20 (8.1) | 1 | | 1 | |
| **Duration** | | | | | | |
| <12 months | 43 (93.5) | 3 (6.5) | 1 | | | |
| 12–60 months | 285 (86.6) | 44 (13.4) | 2.05 (.67–6.34) | 0.213 | | |
| >60 months | 109 (87.9) | 15 (12.1) | 1.86 (0.56–6.11) | 0.310 | | |
| **Opportunistic infection** | | | | | | |
| Yes | 109 (84.5) | 20 (15.5) | 1.366 (0.83–2.24) | 0.216 | | |
| No | 328 (88.6) | 42 (11.4) | 1 | | | |
| **Chronic disease** | | | | | | |
| Yes | 23 (76.7) | 7 (23.3) | 1.99 (0.99–3.99) | 0.052 | 1.28 (0.60–2.74) | 0.519 |
| NO | 414 (88.3) | 55 (11.7) | 1 | | 1 | |
| **BMI category** | | | | | | |
| 18.5–24.9 | 291 (87.9) | 40 (12.1) | 1 | | | |
| <18 | 128 (85.9) | 21 (14.1) | 1.17 (0.71–1.91) | 0.540 | | |
| > = 25 | 18 (94.7) | 1 (5.3) | 0.43 (0.06–3.01) | 0.399 | | |
| **CD4 count category** | | | | | | |
| <200 | 55 (77.5) | 16 (22.5) | **2.40 (1.342–4.29)** | **0.003** | **2.91 (1.58–5.35)** | **0.001** |
| 200–499 | 160 (87.4) | 23 (12.6) | 1.34 (0.77–2.31) | 0.295 | 1.12 (0.64–1.98) | 0.684 |
| > = 500 | 222 (90.6) | 23 (9.4) | 1 | | 1 | |

WHO = World Health Organization, ZDV = zidovudine, BMI = body mass index, PR = Prevalence ratio, CI = Confidence interval, 1.00 = reference group.

(6.5%) [42], Kaduna State, Nigeria (9%) [31], and Ranchi, India (3%) [43]. The observed difference in the prevalence might be due to variation in study populations, clinical conditions, study design methods, and leukopenia definition. In the present study, leukopenia were significantly increased as the CD4 T cell count decreased and age of HIV infected patients increased, which is in agreement with other studies [15, 19, 38]. This could be due to HIV mediated hematopoietic inhibition and direct infection of T cells [4].

Our study showed that the overall prevalence of thrombocytopenia was 12.4% (95% CI; 9.6% - 15.4%). It is consistent with other reports conducted in South West Region of Cameroon (14%) [32] and Southwestern Uganda (13%) [44]. The possible causes of thrombocytopenia could be immune-mediated platelet destruction, impaired platelet production by the infected magakaryocytes of the bone marrow, or myelosuppression effects of medications. On the other hand, the prevalence of this study was higher than studies done in northwest Ethiopia (4.1% and 6.3%) [15, 17], Jimma (6.9%) [19], Addis Ababa (5.7%) [45], and Yaoundé, Cameroon (6.9%) [46], and lower than studies in Ghana (18.5%) [42] and Kaduna State, Nigeria (24%) [26, 31]. The difference might be due to variation in the definition of thrombocytopenia, study design and size of the study population. Regarding the severity of thrombocytopenia, the predominant form was mild thrombocytopenia which is consistent with other studies [45, 46].

Similar to our finding, different studies reported thrombocytopenia were increased and independently associated with degree of immunosuppression [15, 31, 45]. However, our finding deviated from a study conducted in Jimma [19] that showed thrombocytopenia is not associated with neither the degree of immunosuppression nor with the clinical stage of HIV.

Pancytopenia in this study was 1% (95% CI: 0.4%-2%), which is nearly in agreement with a study conducted by Firnhaber C et al (0.3%) [47]. However, a study from Jimma reports no pancytopenia found [19] and studies conducted by Bukar A et al [31] and Santiago-Rodríguez EJ et al [48] reported 8% and 8.7% pancytopenia, respectively, which is higher than our finding. The difference might be due to variation in sample size and design of the studies.

The main limitation of this study is the cross-sectional nature of the study design which does not reveal causal links between independent variables and cytopenias, so a longitudinal study is recommended to generalize the related outcomes of this study. Despite the limitations, the study has determined the magnitude of cytopenias and identified important factors associated with cytopenias in HIV patients on HAART.

In conclusion, in this study prevalence of at least one form of cytopenia was 40% among HIV infected patients on HAART. The most frequent type of cytopenias was anemia followed by leukopenia and thrombocytopenia. Older age group (>50 years old), male gender, ZDV based ART regimen, and lower CD4 T cell count were identified as independent factors associated with having cytopenias (anemia, leucopenia or thrombocytopenia). Therefore, these warrant the need for monitoring hematological parameters of HIV infected patients on HAART to reduce morbidity and mortality.

## Supporting information

**S1 Dataset.**
(XLSX)

## Acknowledgments

We would like to acknowledge Mahal-Meda Hospital staffs for the support given to undertake this study. Our special thanks and appreciation also goes to all study participants for their voluntary participation in the study.

## Author Contributions

**Conceptualization:** Angesom Gebreweld, Daniel Gebretsadik.

**Data curation:** Angesom Gebreweld, Temesgen Fiseha, Nibret Girma, Haftay Haileslasie, Daniel Gebretsadik.

**Formal analysis:** Angesom Gebreweld, Daniel Gebretsadik.

**Investigation:** Angesom Gebreweld, Temesgen Fiseha, Nibret Girma, Haftay Haileslasie, Daniel Gebretsadik.

**Methodology:** Angesom Gebreweld, Temesgen Fiseha, Nibret Girma, Haftay Haileslasie, Daniel Gebretsadik.

**Project administration:** Angesom Gebreweld, Daniel Gebretsadik.

**Resources:** Angesom Gebreweld, Temesgen Fiseha, Nibret Girma, Haftay Haileslasie, Daniel Gebretsadik.

**Software:** Angesom Gebreweld, Temesgen Fiseha, Haftay Haileslasie, Daniel Gebretsadik.

**Supervision:** Angesom Gebreweld, Nibret Girma, Daniel Gebretsadik.

**Validation:** Angesom Gebreweld, Temesgen Fiseha, Haftay Haileslasie.

**Visualization:** Angesom Gebreweld, Temesgen Fiseha.

**Writing – original draft:** Angesom Gebreweld.

**Writing – review & editing:** Angesom Gebreweld, Temesgen Fiseha, Nibret Girma, Haftay Haileslasie, Daniel Gebretsadik.

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
