## [Decision Letter · Decision Letter 0]

21 Apr 2020

PONE-D-20-04588

Prevalence of cytopenia and its associated factors among HIV infected adults on highly active antiretroviral therapy at Mehal Meda Hospital, North Shewa Zone, Ethiopia.

PLOS ONE

Dear Mr Gebreweld,

Thank you for submitting your manuscript to PLOS ONE. After careful consideration, we feel that it has merit but does not fully meet PLOS ONE’s publication criteria as it currently stands. Therefore, we invite you to submit a revised version of the manuscript that addresses the points raised during the review process.

We would appreciate receiving your revised manuscript by Jun 05 2020 11:59PM. To enhance the reproducibility of your results, we recommend that if applicable you deposit your laboratory protocols in protocols.io, where a protocol can be assigned its own identifier (DOI) such that it can be cited independently in the future. For instructions see: http://journals.plos.org/plosone/s/submission-guidelines#loc-laboratory-protocols

We look forward to receiving your revised manuscript.

Kind regards,

Benn Sartorius, PhD

Academic Editor

PLOS ONE

Additional Editor Comments:

Please ensure that the revised manuscript is sent for a full, professional English edit.

Please include as a supplementary file for the revised manuscript a completed STROBE checklist.

Reviewers' comments:

Reviewer's Responses to Questions

**Comments to the Author**

1. Is the manuscript technically sound, and do the data support the conclusions?

Reviewer #1: No

Reviewer #2: Partly

2. Has the statistical analysis been performed appropriately and rigorously? 

Reviewer #1: No

Reviewer #2: No

3. Have the authors made all data underlying the findings in their manuscript fully available?

Reviewer #1: Yes

Reviewer #2: Yes

4. Is the manuscript presented in an intelligible fashion and written in standard English?

Reviewer #1: No

Reviewer #2: No

5. Review Comments to the Author

Reviewer #1: The authors may need input from a native English speaker and the manuscript needs extensive editing for correction of grammatical and other errors for instance from the introduction alone lines 21, 35, 39, 40, 55,57,59,63,68,76-78, 83-85 have grammatical errors.

Introduction

Lines 64-65 China needs to be capitalized but also note that it is part of the industrialized countries. There is wrongful capitalization of words under "Operational definitions" from line 118-130.

Line 88-89: Please mention whether this hospital is located in an urban or rural settings as readers can not deduce this. Table 4 appears to imply that both contexts existed where the study took place and this makes interpretation of the findings difficult. It is sufficient to say "A cross-sectional study was conducted..." and drop the health facility based since the site being a hospital already implies a health facility.

Line 52: It is important for the authors to clearly point out at this point that the causes of cytopenia in HIV infection are multifactorial; HIV infection itself is just one of them but there are other causes of these abnormalities. This is relevant because effective HAART does not reverse all cytopenias

Line 104: No need to describe how BMI is calculated as this is general knowledge. Similarly, it's not relevant to this study to know the expertise of the phlebotomist.

Line 119: Should state clearly that severity of anemia was similarly graded as per WHO rather than state that anemia was further categorized. It is not simply categorizing but rather grading the severity because this has implications at the clinical as well as public health level.

Line 120 has a missing decimal point on 109 g/dl(This is important to correct!)

Line 132: Statistical Analysis

a)The use of the Chi square test in this context is erroneous. This test only provides us with information to the extent that categorical variables in the same population are independent of each other or have a statistical relationship. It is not a measure of the *strength* of an association.

b) The use of Odd Ratios (OR) and logistic regression in a cross-sectional study where the prevalence is high is problematic because it may lead to misinterpretation of the findings by over/underestimating the measure of association between the exposure and outcome. Secondly when OR is used in a cross-sectional study, the language used to report the findings should not use such wording as "higher or more at risk of" (e.g line 184) because the design is such that exposure and outcome are measured simultaneously and therefore can not evaluate risk. The Prevalence Ratio (PR) is recommended in cross-sectional studies with outcomes that have a high prevalence (generally >10%), together with the log-binomial regression model rather than the logistic regression model which may overestimate the measure of association but also give wider confidence intervals.

With that background it would require revising all the tables and reviewing the results and conclusions again. However, below are some additional general comments on the tables

1. Where possible it would be preferable to further combine the categories of the independent variables for instance Marital status: married or unmarried; employment status: employed versus unemployed, ART regimen:Zidovudine (AZT)-based versus non-AZT based and to specify that the duration is with respect to HAART use.

2. Table 3: Once you provide the n for "Yes "and the frequency(percentage) in each category, then the "No" becomes redundant. The "No" column is particularly difficult to interpret for the section of the table showing severity of anemia and thrombocytopenia for instance the row percentages for mild, moderate and severe anemia are difficult to interpret here.

Check typos as well here and throughout all tables.

Discussion:

1. This section should be revised once the issues with the statistical techniques are addressed however, line 283 defeats the purpose of the study. This is the golden opportunity to compare the findings of this study with studies done elsewhere-in Africa, Europe, Asia etc. It is the essence of research! Do my results compare with another study or are they different? What could explain the similarity or differences? Is my population older, more malnourished, more advanced in HIV, and so on. Do my results compare with other parts of Ethiopia or not? What is distinct about the population in Northern Shewa that could explain these findings?

2. Percentages reported need to be rounded of to the nearest whole with appropriate language such as "About 8% of the anemic subjects has severe anemia" instead of 8.62%

3. Line 286 contradicts line 291 with regard to the findings of this study and the one done in Gondar.

4. Lines 285-289: Herein lies the opportunity to give mileage to the importance of this study because you have findings from different parts of Ethiopia especially with regard to prevalence of anemia. Please explore this more and highlight it!

Conclusion: This section too should be revised once the statistical analysis is revised.

Line 343: How high is high considering that some studies have reported figures as high as 80%? What magnitude of abnormality is considered relevant at a public health or global health level? It would be more prudent to conclude within the context of your study for instance, "Anemia was the most common cytopenia (%) in HIV-infected adults receving HAART at the Mahal Meda hospital in Ethiopia. This finding however did not meet the WHO criteria for a severe public health problem"

Reviewer #2: This is an interesting paper describing the prevalence of cytopenias in an HIV population on HAART at a center in Ethiopia. However, substantial revision is necessary.

- This paper would benefit from thorough proof-reading for grammatical (e.g. subject-verb agreement, missing pronous, non-standard capitalization), typographical and spelling mistakes (e.g. "Privet" instead of "private" which are quite extensive throughout the manuscript

- In the tables presenting the main analyses (4,5 and 6), some of the data for the multivariate analyses is missing and it is not apparent why this is the case.

- In table 3, I would suggest using the words "prevalence" or "frequency" instead of "magnitude". Also, the definitions of "mild, moderate, severe" should be included either in the table or as a footnote to the table

-For Table 1, consider rearranging the table or selecting the appropriate presentation of the data, as some of the variables are summarized twice

-It was not clear how the sample size of 499 was selected. Was this based on potential precision of the prevalence estimates or the comparative univariate analyses?

- Some discussion of the limitations of the methods and statistical analysis is warranted. For example, most of the outcomes were common (>10%) and logistic regression has a tendency to be biased away from the null when the dependent variable is not rare. Additionally, please comment on whether any interactions between the variables were explored.

-For the result of a higher prevalence of anemia among males, it would be good to explain why that may be the case in this particular patient population given what has previously been observed from studies in the literature

6. PLOS authors have the option to publish the peer review history of their article (what does this mean?). If published, this will include your full peer review and any attached files.

Reviewer #1: No

Reviewer #2: No

---

## [Author Response · Author response to Decision Letter 0]

5 Jun 2020

Response to Academic editor

Response: We have followed PLOS ONE's formatting guidelines

2. Please ensure that the revised manuscript is sent for a full, professional English edit.

Response: The revised manuscript was edited by professional English editor.

3. Please include as a supplementary file for the revised manuscript a completed STROBE checklist.

 Response: The completed STROBE checklist has included as a supplementary file

Response to reviewer #1

We appreciate that the reviewer’s comments. The followings are our point-by-point responses:

Comment: The authors may need input from a native English speaker and the manuscript needs extensive editing for correction of grammatical and other errors for instance from the introduction alone lines 21, 35, 39, 40, 55,57,59,63,68,76-78, 83-85 have grammatical errors.

Response: We tried to correct grammatical, typographical and spelling errors in the manuscript and the revised manuscript was edited by professional English editor. 

Introduction

Lines 64-65 China needs to be capitalized but also note that it is part of the industrialized countries. There is wrongful capitalization of words under "Operational definitions" from line 118-130.

Response: We corrected it (see line 65 and from 118-133)

Line 88-89: Please mention whether this hospital is located in an urban or rural settings as readers cannot deduce this. Table 4 appears to imply that both contexts existed where the study took place and this makes interpretation of the findings difficult. 

Response: As we stated in methodology section, the hospital is found in Mehal Meda town and provide service for both urban and rural residents (see line 91). 

It is sufficient to say "A cross-sectional study was conducted..." and drop the health facility based since the site being a hospital already implies a health facility.

Response: We have already dropped it. (See line 88)

Line 52: It is important for the authors to clearly point out at this point that the causes of cytopenia in HIV infection are multifactorial; HIV infection itself is just one of them but there are other causes of these abnormalities. This is relevant because effective HAART does not reverse all cytopenias

Response: We have added the possible causes of cytopenia in HIV infection. (Line 52-55)

Line 104: No need to describe how BMI is calculated as this is general knowledge. Similarly, it's not relevant to this study to know the expertise of the phlebotomist.

Response: We have removed it (line 106).

Line 119: Should state clearly that severity of anemia was similarly graded as per WHO rather than state that anemia was further categorized. It is not simply categorizing but rather grading the severity because this has implications at the clinical as well as public health level.

Response: We have stated it clearly, anemia was categorized using the WHO classification criteria (see line 120).

Line 120 has a missing decimal point on 109 g/dl(This is important to correct!)

Response: We corrected it (see line 122).

Line 132: Statistical Analysis

a) The use of the Chi square test in this context is erroneous. This test only provides us with information to the extent that categorical variables in the same population are independent of each other or have a statistical relationship. It is not a measure of the *strength* of an association.

Response: We have corrected the statement (see line 138). 

b) The use of Odd Ratios (OR) and logistic regression in a cross-sectional study where the prevalence is high is problematic because it may lead to misinterpretation of the findings by over/underestimating the measure of association between the exposure and outcome. Secondly when OR is used in a cross-sectional study, the language used to report the findings should not use such wording as "higher or more at risk of" (e.g line 184) because the design is such that exposure and outcome are measured simultaneously and therefore cannot evaluate risk. The Prevalence Ratio (PR) is recommended in cross-sectional studies with outcomes that have a high prevalence (generally >10%), together with the log-binomial regression model rather than the logistic regression model which may overestimate the measure of association but also give wider confidence intervals. With that background it would require revising all the tables and reviewing the results and conclusions again.

Response: Based on the recommendation, instead of odds ratio, we have estimated the Prevalence Ratio (PR) and its 95% confidence interval as a measure of association between the exposure and outcome using Poisson regression model with robust error variance (see from line 139-144). We couldn’t use log-binomial regression model because convergence problems. We revised all the tables, results, and conclusions. 

Below are some additional general comments on the tables

1. Where possible it would be preferable to further combine the categories of the independent variables for instance Marital status: married or unmarried; employment status: employed versus unemployed, ART regimen: Zidovudine (AZT)-based versus non-AZT based and to specify that the duration is with respect to HAART use.

Response: We merged some categories of the independent variables for instance marital status: married or unmarried; employment status: employed versus unemployed, ART regimen: Zidovudine (AZT)-based versus non-AZT based (see tables).

2. Table 3: Once you provide the n for "Yes "and the frequency (percentage) in each category, then the "No" becomes redundant. The "No" column is particularly difficult to interpret for the section of the table showing severity of anemia and thrombocytopenia for instance the row percentages for mild, moderate and severe anemia are difficult to interpret here. Check typos as well here and throughout all tables.

Response: We omitted the ‘’No’’ column from table 3 to reduce redundancy. We have corrected the typographical errors. 

Discussion:

1. This section should be revised once the issues with the statistical techniques are addressed however, line 283 defeats the purpose of the study. This is the golden opportunity to compare the findings of this study with studies done elsewhere-in Africa, Europe, Asia etc. It is the essence of research! Do my results compare with another study or are they different? What could explain the similarity or differences? Is my population older, more malnourished, more advanced in HIV, and so on. Do my results compare with other parts of Ethiopia or not? What is distinct about the population in Northern Shewa that could explain these findings?

Response: We have revised the discussion part and our finding is compared with other studies.

2. Percentages reported need to be rounded of to the nearest whole with appropriate language such as "About 8% of the anemic subjects has severe anemia" instead of 8.62%

Response: We rounded off the percentages to the nearest whole number (see line 289)

3. Line 286 contradicts line 291 with regard to the findings of this study and the one done in Gondar.

Response: line 286 not contradict with line 291 because they are two different studies conducted in Gondar by different authors. 

4. Lines 285-289: Herein lies the opportunity to give mileage to the importance of this study because you have findings from different parts of Ethiopia especially with regard to prevalence of anemia. Please explore this more and highlight it!

Response:

Conclusion: This section too should be revised once the statistical analysis is revised.

Line 343: How high is high considering that some studies have reported figures as high as 80%? What magnitude of abnormality is considered relevant at a public health or global health level? It would be more prudent to conclude within the context of your study for instance, "Anemia was the most common cytopenia (%) in HIV-infected adults receiving HAART at the Mahal Meda hospital in Ethiopia. This finding however did not meet the WHO criteria for a severe public health problem"

Response: We have revised the conclusion section and substituted the word ‘’high’’ by the actual magnitude (see line from 341-347). 

Response to reviewer #2

We appreciate that the reviewer’s comments. The followings are our point-by-point responses:

- This paper would benefit from thorough proof-reading for grammatical (e.g. subject-verb agreement, missing pronouns, non-standard capitalization), typographical and spelling mistakes (e.g. "Privet" instead of "private" which are quite extensive throughout the manuscript

Response: We tried to correct grammatical, typographical and spelling errors in the manuscript and the revised manuscript was edited by professional English editor.

- In the tables presenting the main analyses (4,5 and 6), some of the data for the multivariate analyses is missing and it is not apparent why this is the case..

Response: Variables with p < 0.20 at bivariate analysis were included in the multivariable analysis that is why some of the data for the multivariate analyses are missed. (see from line142-144)

- In table 3, I would suggest using the words "prevalence" or "frequency" instead of "magnitude". Also, the definitions of "mild, moderate, severe" should be included either in the table or as a footnote to the table

Response: based on the suggestion, we replaced the word "magnitude" by prevalence (see table 3). We have already stated the definitions of severity of anemia and thrombocytopenia (mild, moderate, severe) in method section under operational definitions. To reduce redundancy we haven’t included in the table or as a footnote to the table. (See from line 120-123 and line 127-129)

-It was not clear how the sample size of 499 was selected. Was this based on potential precision of the prevalence estimates or the comparative univariate analyses?

Response: We tried to make clear how the sample size of 499 was selected (line 96-100). 

- Some discussion of the limitations of the methods and statistical analysis is warranted. For example, most of the outcomes were common (>10%) and logistic regression has a tendency to be biased away from the null when the dependent variable is not rare. Additionally, please comment on whether any interactions between the variables were explored.

Response: We have included some limitations of the study (see from line 336-340). However, we revised the statistical analysis we had used before. Instead of odds ratio, we have estimated the Prevalence Ratio (PR) and its 95% confidence interval as a measure of association between the exposure and outcome using Poisson regression model with robust error variance. So we didn’t included as limitation of the study (see line 139-144).

-For the result of a higher prevalence of anemia among males, it would be good to explain why that may be the case in this particular patient population given what has previously been observed from studies in the literature

Response: In our study, the prevalence of anemia was higher in males than females; this might be due to the difference in the definition of anemia (see line 294-296). However, it needs further study to know the exact reason in the study area. 

-For Table 1, consider rearranging the table or selecting the appropriate presentation of the data, as some of the variables are summarized twice

Response: We have rearranged table 1 by merging some categories of the independent variables.

---

## [Decision Letter · Decision Letter 1]

6 Aug 2020

PONE-D-20-04588R1

Prevalence of cytopenia and its associated factors among HIV infected adults on highly active antiretroviral therapy at Mehal Meda Hospital, North Shewa Zone, Ethiopia.

PLOS ONE

Dear Dr. Gebreweld,

Thank you for submitting your manuscript to PLOS ONE. After careful consideration, we feel that it has merit but does not fully meet PLOS ONE’s publication criteria as it currently stands. Therefore, we invite you to submit a revised version of the manuscript that addresses the points raised during the review process.

We look forward to receiving your revised manuscript.

Kind regards,

Benn Sartorius, PhD

Academic Editor

PLOS ONE

Reviewers' comments:

Reviewer's Responses to Questions

**Comments to the Author**

1. If the authors have adequately addressed your comments raised in a previous round of review and you feel that this manuscript is now acceptable for publication, you may indicate that here to bypass the “Comments to the Author” section, enter your conflict of interest statement in the “Confidential to Editor” section, and submit your "Accept" recommendation.

Reviewer #2: (No Response)

2. Is the manuscript technically sound, and do the data support the conclusions?

Reviewer #2: Yes

3. Has the statistical analysis been performed appropriately and rigorously? 

Reviewer #2: Yes

4. Have the authors made all data underlying the findings in their manuscript fully available?

Reviewer #2: Yes

5. Is the manuscript presented in an intelligible fashion and written in standard English?

Reviewer #2: Yes

6. Review Comments to the Author

Reviewer #2: The authors have addressed most of my concerns. Although substantially improved from the first draft, the manuscript still requires additional thorough proof-reading and correction of grammatical/typographic errors such as consistency in verb tense, non-standard capitalization (e.g. "Male" instead of "male" in the middle of a sentence see line 256 as an example), missing words (e.g. see line 232 "Anemia *was* also...").

Additionally, the tables also require some revisions. One, each table should be able to stand alone and not rely on the text, hence the definitions of severity should be included in the footnote. Secondly, the standard number of decimal places for effect estimates (e.g. PR) is 2. In the tables the number of decimal places is inconsistent (sometimes 2 and sometimes 3). Additionally, there should be a space between the n and the %, e.g. 128 (66.0) instead of 128(66.0).

7. PLOS authors have the option to publish the peer review history of their article (what does this mean?). If published, this will include your full peer review and any attached files.

Reviewer #2: No

---

## [Author Response · Author response to Decision Letter 1]

18 Aug 2020

Response to reviewer 

We appreciate the reviewer’s comments. The followings are our point-by-point responses:

Comment: The authors have addressed most of my concerns. Although substantially improved from the first draft, the manuscript still requires additional thorough proof-reading and correction of grammatical/typographic errors such as consistency in verb tense, non-standard capitalization (e.g. "Male" instead of "male" in the middle of a sentence see line 256 as an example), missing words (e.g. see line 232 "Anemia *was* also..."). 

Response: We have corrected the grammatical/typographic errors, non-standard capitalizations, and missing words in the manuscript.

Comment: Additionally, the tables also require some revisions. One, each table should be able to stand alone and not rely on the text, hence the definitions of severity should be included in the footnote. Secondly, the standard number of decimal places for effect estimates (e.g. PR) is 2. In the tables the number of decimal places is inconsistent (sometimes 2 and sometimes 3). Additionally, there should be a space between the n and the %, e.g. 128 (66.0) instead of 128(66.0).

. 

Response: We have included the definitions of severity in the footnote of Table 3.We have corrected the number of decimal places for PR in the tables and texts and we have put a space between n and % in the tables.

Looking forward to hearing from you. Thank you again for your consideration! 

Sincerely, 

Angesom Gebreweld (BSc, MSc)

---

## [Editor Report · Decision Letter 2]

2 Sep 2020

Prevalence of cytopenia and its associated factors among HIV infected adults on highly active antiretroviral therapy at Mehal Meda Hospital, North Shewa Zone, Ethiopia.

PONE-D-20-04588R2

Dear Dr. Gebreweld,

We’re pleased to inform you that your manuscript has been judged scientifically suitable for publication and will be formally accepted for publication once it meets all outstanding technical requirements.

Kind regards,

Benn Sartorius, PhD

Academic Editor

PLOS ONE
---

## [Editor Report · Acceptance letter]

7 Sep 2020

PONE-D-20-04588R2 

Prevalence of cytopenia and its associated factors among HIV infected adults on highly active antiretroviral therapy at Mehal Meda Hospital, North Shewa Zone, Ethiopia. 

Dear Dr. Gebreweld:

I'm pleased to inform you that your manuscript has been deemed suitable for publication in PLOS ONE. Congratulations! Your manuscript is now with our production department. 

Kind regards, 

on behalf of

Dr. Benn Sartorius 

Academic Editor

PLOS ONE